# Potential Therapeutic Effects of Thiazolidinedione on Malignant Glioma

**DOI:** 10.3390/ijms232113510

**Published:** 2022-11-04

**Authors:** Meei-Ling Sheu, Liang-Yi Pan, Huai-Yun Hu, Hong-Lin Su, Jason Sheehan, Hsi-Kai Tsou, Hung-Chuan Pan

**Affiliations:** 1Institute of Biomedical Sciences, National Chung-Hsing University, Taichung 402, Taiwan; 2Faculty of Medicine, Kaohsiung Medical University, Kaohsiung 80708, Taiwan; 3Department of Life Sciences, Agriculture Biotechnology Center, National Chun-Hsing University, Taichung 402, Taiwan; 4Department of Neurosurgery, University of Virginia, Charlottesville, VA 22908, USA; 5Department of Neurosurgery, Taichung Veterans General Hospital, Taichung 40210, Taiwan; 6Department of Medical Research, Taichung Veterans General Hospital, Taichung 40210, Taiwan

**Keywords:** thiazolidinedione, PPARγ, glioma, STAT3, SHP-2

## Abstract

Glioblastoma multiforme (GBM) is the most common and aggressive primary malignant tumor of the central nervous system. GBM has a very low 5-year survival rate and reaching merely a median of ~15 months even with aggressive treatments. PPARγ (Peroxisome proliferator- activated receptor gamma) agonists (ciglitazone), while being widely used on patients of type 2 diabetes mellitus, also have approved anticancer effects. Their action mechanisms on malignant glioma are not fully understood. The aim of this study is to investigate the potential therapeutic effect of PPARγ agonists on maligant glioma. Glioma cell line and in-vivo/ex-vivo animal model intervened by ciglitazone were used to assess the associated mechanism and therapeutic effect. Our results from in vivo and ex vivo experiments showed that ciglitazone not only inhibited tumor growth and its associated angiogenesis, but it also reduced colony formation and migration of tumors. Ciglitazone inhibited the phosphorylation of STAT3 (signal transducer and activator of transcription 3) (at the point of tyrosine 705 by increasing both the amount and activity of SHP-2 (Src homology region 2-containing protein tyrosine phosphatase 2) proteins, based on evidence obtained from immunoprecipitation and immunohistochemistry. Furthermore, ciglitazone activated proteasomes and lysosomes to degrade cell-cycle-related proteins like Cyclin D1, Cyclin E, CDK2 (Cyclin-dependent kinase 2), and CDK4 (Cyclin-dependent kinase 4). Ciglitazone triggered expressions of LC3 (Microtubule-associated protein 1A/1B-light chain 3) and formation of acidic vesicular organelles (AVOs), both of which were implicated in the autophagy pathway. In conclusion, ciglitazone showed the multiple actions to regulate the growth of glioma, which appeared to be a potential candidate for treating malignant glioma.

## 1. Introduction

Glioblastoma multiforme (GBM) is the most aggressive primary malignancy of the central nervous system with a <5% 5-year survival rate [1]. Even with aggressive treatment strategies consisting of maximum safe resection, adjuvant chemoradiotherapy, and target therapy, the median overall survival is only 15 months [2]. Despite technological and therapeutic advances, like inhibition of oncogenic signal transduction, antiangiogenesis and immunotherapy, GBM remains incurable [3]. New novel targets for GBM treatment are therefore in demand.

PPARγ regulates fat metabolism, energy homeostasis, proliferation, and inflammation, and its presence is found in a variety of cancer cells, including glioblastoma [4]. PPARγ ligands were originally characterized as having an antidiabetic effect. Other evidence has shown that one of such ligands, rosiglitazone, inhibits proliferation of glioblastoma cell lines, like U87-MG, by arresting cell cycle at G2/M, cell cycle and promoting apoptosis. The growth inhibitory effect was partially reversed by the PPARγ antagonist GW9662, suggesting that rosiglitazone most likely acts through a PPARγ-dependent pathway [4]. Another PPARγ synthetic ligand, ciglitazone, produces a rapid dose-dependent loss of mitochondrial membrane potential, along with a rapid overproduction of reactive oxygen species (ROS) in rat glioma cells [5]. In clinical assessment, diabetic patients with glioblastoma treated with PPARγ agonists showed a longer median survival of 19 months, compared to 6 months in similar patients receiving the standard GBM treatment [6] PPARγ ligands could therefore be a potential drug for the treatment of glioblastoma [7].

Several transcriptional pathways are likely targeted by PPARγ ligands. For example, STAT3 is a crucial transcription factor regulating immune response [8] and tumorigenesis [9]. STAT3 is activated in glioblastoma tumors and glioblastoma-derived cell lines [10]. Phosphorylation of STAT3 is associated with poor clinical outcome of glioblastoma patients [11]. In glioblastoma spheres, PPARγ ligands induce cell cycle arrest and apoptosis together with the inhibition on the STAT3 pathway [12]. STAT3 modulates VEGF (vascular endothelial growth factor) expression and cell migration of the endothelium, contributing to angiogenesis. These effects are abolished by the downregulation of STAT3 [13,14,15]. The phosphorylation of STAT3 is also highly expressed in the endothelium of GBM, involving angiogenesis [16]. Thus, the attenuation of angiogenesis by PPARγ ligands in GBM is likely mediated through STAT3 cascades. 

The modulation of cell cycle is crucial for tumor development mostly through the interaction of cyclin and cyclin-dependent kinase [17]. The invasiveness and malignancy of GBM are highly correlated with mutation at the p16INK4a/ARF locus, P53, chromosome 9p deletions, and inactivation of the Rb gene [18,19]. High expressions of CDK4 also contribute to the degree of malignancy [20]. PPARγ ligands activate cyclin-dependent kinase inhibitors of P21 and P27, and they also inhibit cyclin D1 to arrest cell cycles in pancreatic carcinoma and hepatoma [21]. In addition, PPARγ induces expressions of Bax and Bad, releasing cytochrome C, and it activates caspase 3 to produce apoptosis of the above malignancies [22,23]. PPARγ therefore could serve as a potential target to modulate cell cycle to arrest the tumor growth in GBM. 

Ciglitazone (thiazolidinedione analogs) is a well-known PPARγ ligand in treating diabetes with the potential for tumor treatment [24,25]. However, the role of PPARγ ligand in treating GBM especially with regard to angiogenesis and cell cycle regulation is not fully understood. Here, the aim of this study is to investigate the effects of ciglitazone, a PPARγ ligand, on GBM, both in vivo and in vitro, to assess its potential clinical applications. 

## 2. Results

### 2.1. Arrest of Glioma Growth by Ciglitazone Either In Vitro and Vivo Assessment

U87 cell lines at cell density of 2 × 10^4^ cells/mL were plated and subjected to individual treatments of 20 μM ciglitazone, 20 μM troglitazone, or 10 μM 15d-PGJ_2_. Photographs of the soft agar assay showed fewer colony formation when subjected to treatment of either ciglitazone, troglitazone, or 15d-PGJ2 (Figure 1A). Giemsa staining also showed statistically reduced colony formation both in terms of morphology (Figure 1B) and quantitative analysis (Figure 1C). 

The wound healing test is essential for the assessment of tumor cell migration. U87 cells were incubated with PPARγ agonists consisting of either ciglitazone, troglitazone, or 15d-PGJ2 at the concentration of 5 μM for 48 h. The above-said PPARγ have the significant power to inhibit U87 cell migration (Figure 1D,E). 

Subcutaneous injections of U87 cells at an amount of 5 × 10^6^ cells were assessed when the tumor had grown to 5 mm in diameter. That was used as the baseline for further comparisons. The intratumor injection of 2 mg/kg ciglitazone or 2 mg/kg 15d-PGJ2 inhibited the tumor growth as measured in photographs taken from the flank region of the animal (Figure 2A). After isolating the tumor, we measured its size (Figure 2B) and weight (Figure 2C), which showed a significant decrease compared to the control.

### 2.2. Antiangiogenesis Effects of Ciglitazone on Malignant Glioma 

The matrix metalloproteins (MMP) play a critical role in tumor angiogenesis and metastasis. U87 cells were treated with ciglitazone at graded concentrations of 5, 10, 20, 30, and 40 μM. Reduced activities of MMP-2 and MMP-9 were highly correlated with concentrations of ciglitazone (Figure 3A). 

New growth of vascular structure is essential for the tumor development especially for sizes >1 to 2 mm. To assess ciglitazone effects on vessel growth, we conducted the aorta ring test. At 5 or 20 μM treatment, ciglitazone showed inhibition of the endothelial growth (Figure 3B). In the subcutaneous Matrigel plug assay, either VEGF (4.28 ± 0.84 lumen/per field) (*p* < 0.01) or bFGF (4.14 ± 1.33 lumen/per field) (*p* < 0.01) triggered the significant vascular formation compared to control (0.14 ± 0.11 lumen/per field). Combined with the endothelial growth supplement, further effect was found in VEGF + ECGS (19.4 ± 4.2 lumen/per field) (*p* < 0.001) as a positive control. Angiogenesis was abolished by ciglitazone injections in VEGF+ ciglitazone (1.1 ± 0.61 lumen/per field) (*p* < 0.01) and bFGF+ ciglitazone (0.85 ± 0.593 lumen/per field) (*p* < 0.01) (Figure 3C). 

### 2.3. Abolishment of Glioma Growth by Ciglitazone through STAT3 Pathway in Both In Vitro and In Vivo Experiments

Over 90% of GBM showed high expressions of STAT3 and phosphorylation of STAT3 harbored the worst survival [26]. We further assessed the role of PPARγ in suppressing tumor growth through modulation of STAT3. U87 cells were subjected to different concentrations of ciglitazone (2.5, 5, 10, and 20 μM) and studied at different time points (15 min, 30 min, 1 h, 2 h, 4 h, 8 h, 12 h, and 24 h). Regardless of different concentrations or time points, ciglitazone consistently suppressed tumor growth through the abolishment of the P-STAT3 (Tyr 705) activity (Figure 4A,B). 

Dephosphorylation of STAT3 at the locus of tyrosine was mediated mainly through protein tyrosine phosphatase (PTP), and SHP-1 and SHP-2 were the major determinants. Results showed no increased expression in SHP-1 in U87 cells treated with 20 μM ciglitazone at different time points. However, at a longer period of time, SHP-2 was more marked (Figure 4C). Treated with sodium stibogluconate (SSG) (PTP inhibitor), low concentration at 11 μM inhibited p-STAT3 activity. When concentration reached 110 μM, p-STAT3 activity became inhibited. Ciglitazone hence induced SHP-2 activity to further exert de-phosphorylation of p-STAT3 in the U87 cell line (Figure 4D). 

### 2.4. Ciglitazone Induced Strong Interactions of p-STAT3 and SHP-2 in U87 Cells

For further assessing the interaction of p-STAT3 and SHP-2, U87 cells were treated with ciglitazone in the different time points of 1, 2, and 4 h and assessed by the immunoprecipitation. The data showed the strong interaction between p-STAT3 (Tyr 705) and SHP-2 especially highest at a 1 h time point (Figure 5A). The immunohistochemistry staining of p-STAT3 and SHP-2 showed high colocalization of the above-said proteins and also highest colocalized at treatment of 1 h (Figure 5B). U87 were treated with vehicle or ciglitazone 20 μM for 1 h, and whole cell lysates for SHP-2 activity assay determined by adding the PNPP substrate showed the significant SHP-2 activity between control and ciglitazone up to 24 h (Figure 5C). Immunohistochemistry analysis for phospho-STAT3 (Tyr 705) and SHP-2 in subcutaneous tumor showed strong expression of SHP-2 and reciprocal decrease in phospho-STAT3 by ciglitazone (Figure 5D). The above data confirmed the high interaction of p-STAT3 and SHP-2 in U87 cells or tumor treated with ciglitazone. 

### 2.5. Glioma Tumor Growths Arrested by Ciglitazone Involved in Cell Cycle Associated with Proteasome and Lysosome Activity 

For the assessment of direct cell cytotoxic effect of ciglitazone to influence glioma cell survival, glioma cell lines (U87, H4, 8401, 8901) were subjected to different concentration of ciglitazone from 10, 20, 30, and 40 μM for 24 h. The above-said cell lines were analyzed by MTT assay, which showed the influence on glioma cell survival by ciglitazone at the concentration of 20 μM (Figure 6A). Another set of U87 cells treated with 20 μM ciglitazone was subjected to propidium iodide staining for the flow cytochemistry evaluation; there was also significant influence in cell survival in U87 cell treated with ciglitazone (Figure 6B,C). The same phenomena was also observed in U87 cells treated with other thiazolodinedione analogs (Troglitazoe and 15d−PGJ2) seen in Appendix A. It is well known that ciglitazone could inhibit the cyclin D1 and cyclin B expression [27]. The U87 cell line was treated with 20 μM ciglitazone, and cells harvested at the different time points. The Western blot analysis revealed that CDK2, CDK4, CDK6, cyclin D1, and cyclin E were inhibited with escalation profile related to different time profile (Figure 6C). The quantitative analysis revealed the significant difference starting at the time points of 8 h lasting to 24 h (Figure 6D). 

### 2.6. Induction of Proteasome and Lysosome Formation by Ciglitazone

The degradations of proteins were mainly through ER stress, proteasome, and lysosome. For the determination of protein degradation either through the above-said pathway, U87 cell lines were treated with 20 μM ciglitazone and subjected to Western blot analysis in the different time profile. The data revealed that there was no significant expression in calpain I and II related to various time points (Figure 7A). However, there was significant expression in ubiquitin and lysosomal-associated membrane protein 1 (LAMP1), and even higher expression was determined at the delayed time points up to 24 h (Figure 7B,C). Thus, the protein degradation was mainly through the induction of proteasome and lysosome, but it was not ER stress.

The U87 cell line treated 20 μM ciglitazone was subjected to proteasome inhibitors (MG 132, Lactacystin) and lysosome inhibitors (bafilomycin, chloroquine diphosphate slat) for 24 h. The results showed the decreased cell cycle proteins (CDK2, CDK4, CDK6, cyclin D1, and Cyclin E) by ciglitazone were restored by proteasome and lysosome inhibitors (Figure 7D,E). The quantitative analysis were shown in Figure 7F–J.

### 2.7. Ciglitazone-Induced Autophagy in U87 Cells through Overexpressing LC3 

Based on the above findings, ciglitazone arrested the cell cycle and changed the subG0/G1 ratio. The involvement of autophagy cascade may be implicated. The alteration from LC3-I to LC3-II is an essential biomarker for the development of autophagosomes. U87 cells treated with 20 μM ciglitazone expressed significant amounts of LC3-II, in a manner dependent on length of time for application (Figure 8A). The phenomenon was reversed by bafilomycin (autophagosome inhibitor) (Figure 8B). In another assay to determine autophagosomes, we found a strong expression of acidic vesicular organelles (AVOs) as revealed by acridine orange staining (Figure 8C). The quantitative analysis were shown in Figure 8D,E. 

## 3. Discussion

We found that ciglitazone had multiple actions on the glioma, including the inhibition of angiogenesis, migration, decreased STAT3 phosphorylation, arrest of the cell cycle, activation of proteasome and lysosome, and increased autophagy. Based on its known drug safety during long-term use in diabetes mellitus patients, ciglitazone application, either local or systemic is likely a potential treatment strategy for malignant glioma. 

STAT3, highly expressed in the multiple malignant tumors, is essential for growth and transformation of tumors [9,28,29,30]. In glioma, STAT3 is also overexpressed, and it is important for tumor cell cycle, tumor growth, and transformation as well as the angiogenesis [31,32]. Hypoxia, necrosis, and angiogenesis are hallmarks of GBM. Under hypoxic conditions, the dominant negative mutant (DN)-STAT3 is highly expressed, thereby exerting a negative effect on tumor growth and angiogenesis [33]. These findings implicated that STAT3 is crucial for GBM growth. In this study, we found that ciglitazone decreased phosphor-STAT3 (Try 705) activities, coupled with the reciprocal increase of SHP-2 in both glioma cell line and glioma tissue. Data are in support of the fact that ciglitazone inhibits phosphor-STAT3 and activates SHP-2 expressions. 

The cell cycle proteins of CDK2, CDK4, and cyclin D1 and cyclin E regulate cells transforming from G1 to S phase [17]. Ciglitazone is known to reduce expressions of both cyclins B and D1 in GBM [34]. In this study, we found that ciglitazone inhibited the expression of CDK2, CDK4, cyclin D1, and cyclin E. Regarding the degradation of the above-said proteins, three major pathways are known, including ER Stress, proteosome, and lysosome activity [35,36,37]. Results showed that ciglitazone activated the expression of Ubiquitin and LAMP1 expressions, and these activations were attenuated by inhibitors. Findings confirmed that the degradation of cell cycle proteins by ciglitazone was mediated through activated proteosomes and lysosomes. Results are compatible with the report that activation of proteosomes in MCF-7 breast cell line is triggered by ciglitazone [38]. 

In a dosage study of ciglitazone on cell death in U87 MG, A172, and C6 glioma cells lines, ciglitazone reached as high as 30 μM to exert significant effects [39]. In another similar study on U87 cells, the IC50 of ciglitazone reached 170 μM, and cell arrest in G1 and G2/M phases is mediated through lowering cyclins D1 and B, while raising P27 and P21 [40]. In our present study, U87 cells treated with ciglitazone up to 20 μM showed the significant difference in cell survival as assessed by MTT and flow cytochemistry, a finding implicating the possible involvement of autophagy. Upon autophagy, LC3-I changes to LC3-II and fusing cytoplasmic membranes form autophagosomes and autolysome [41]. Our data revealed an increased transformation of LC3-I to LC3-II, which is a prerequisite for developing autophagosomes and autolysosomes. The immunohistochemistry in positive acridine orange staining further confirmed the trigger of autophagy of GBM by ciglitazone. 

Rosiglitazone’s inhibition of tumor growth is mediated partly through the inhibition of angiogenesis. Even at low concentrations, rosiglitazone inhibits the proliferation of bovine capillary endothelial cells, as well as VEGF secretions from tumor cells [42]. In our last study, we also found that rosiglitazone inhibits the growth and DNA synthesis as well as tube formation in endothelial cells [43]. In that study, results from the aorta ring assay and Matrigel plug study also showed that rosiglitazone suppresses angiogenesis. Thus, the antiangiogenesis of rosiglitazone could be a possible target for the treatment of malignant glioma.

## 4. Materials and Methods

### 4.1. Subcutaneous U87 MG Xenograft

The U87 MG cell line, at the concentration of 5 × 10^7^ cells in 200 μL of complete medium, was bilaterally injected into flank regions of the animal on both sides. When the tumor reached 5 mm in size, 2 mg/kg ciglitazone (Sigma-Aldrich, St. Louis, MO, USA) or 2 mg/kg 15d-PGJ2 (Sigma-Aldrich, St. Louis, MO, USA) was injected into the tumor two times a week. Tumor was removed 10 days after the last injection and allowed for measurement of tumor size and volume. 

### 4.2. Cell Culture

The U87 MG cell line was obtained with help from Dr Chun-Chung Chen (Department of medical research, Taichung Veterans General Hospital, Taichung, Taiwan). U87 MG cells were cultured in DMEM by adding 10% fetal bovine serum, 2 mM L-glutamine, 100 U/mL Penicillin, 100 μg/mL Streptomycin, 0.25 μg/mL Amphotericin B, and 110 mg/L Sodium pyruvate, and were incubated at 5% CO_2_ at 37 °C. When the U87 MG cells reached near 80% of confluence, they were passed to another cell culture dish for further study. U87 cells were dissociated with Trypsin-EDTA and cultured in DMEM before spun at 1200 rpm for 5 min for the next study. 

### 4.3. Matrigel Plug Assay

Nude mice or C57BL/6 mice were injected subcutaneously with 0.2 mL of Matrigel containing the desired growth factors or chemicals, and animals were separated into various experimental groups. These injected chemicals and the associated groups were VEGF(Calbiochem, San Diego, CA, USA)+ ECGS (BD Biosciences, Bedford, MA), VEGF, VEGF+ ciglitazone, bFGF (Sigma-Aldrich,. St. Louis, MO, USA), and bFGF+ ciglitazone. The injected Matrigel rapidly formed a single, solid gel plug. After 14 days, the skin of the mouse was pulled back to expose the Matrigel plug, which remained intact and subjected to hematoxylin and eosin stainings. The stained images were photographed with a Nikon digital camera, and newly formed microvessels were counted.

### 4.4. Ex Vivo Vessel Sprouting Aortic Ring Assay

Aortas were isolated from 6-week-old Sprague-Dawley rats. Plates (48-well) were coated with 120 μL of Matrigel. Periadventitial fat and connective tissues were cleaned off from isolated aortas and cut into 1- to 1.5 mm-long rings. After rinsing 5 times with the endothelial cell–based medium, aortas were placed on the Matrigel-coated wells and covered with another 100 μL of Matrigel. Five μM and 20 μM ciglitazone were added each to a final 250 μL of medium. Photographs were taken at days 0, 1, and 3. Cultures were incubated, with the medium replaced every other day throughout the course of 8 to 10 days of experiments. Visual counts of microvessel outgrowths from replicate explant cultures (*n* = 6) were carried out under bright-field microscope following an established protocol. Experiments were repeated at least 4 times, and microvessel counts in treated and control cultures were compared.

### 4.5. Immunoprecipitations Analyses 

Proteins (80 μg) were separated by sodium dodecyl sulfate–polyacrylamide gel electrophoresis and electrophoretically transferred to nitrocellulose membranes. The end-product was blocked for 1 h in PBS containing Tween 20 (0.1%) and nonfat milk (5%). Blots were incubated with p-STAT 3 and SHP-2 for 1 h. Membranes were then incubated for 1 h together with HRP-conjugated secondary antibodies. After PBS washing, blots were incubated with commercial chemiluminescence reagents (Amersham Biosciences, Piscataway, NJ). The quantitative analysis of the protein expression was determined with a densitometer (Image-Pro Plus software, Version 6.0).

### 4.6. Western Blot Analyses

After different treatments, U87 MG cells were harvested and added to the RIPA lysis buffer for several minutes before being stored at −20 °C overnight. The lysates were centrifuged at 14,000 rpm at 4 °C for 10 min. The supernatant was collected for protein analysis. Total proteins were extracted, resolved by SDS-polyacrylamide gel electrophoresis, and transferred onto a blotting membrane. After blocking with nonfat milk, membranes were incubated with primary antibodies (Appendix A). Membranes were incubated with the horseradish peroxidase-conjugated secondary antibody and developed using ECL Western blotting reagents. The intensity of the protein bands was determined through a computerized image analysis system (IS1000, Alpha Innotech Corporation, Santa Clara, CA, USA). 

### 4.7. Protein Tyrosine Phosphatase Activity Assay

U87 cells were seeded on a 10 cm size culture dish and subjected to different treatments. Supernatants were discarded, and the culture dish was washed twice with PBS before being stored at −20 °C with additional Baco IP buffer overnight. Centrifugation was at 14,000 rpm, 4 °C for 10 min. The supernatant of 1 gm was incubated with 10 μL SHP-2 antibody at 4 °C overnight and then 35 μL protein A Sepharose was added and incubated for another day. Finally, it was centrifuged at 1200 rpm, 4 °C for 10 min. The supernatant was discarded and washed with Baco IP buffer before it was incubated for 1 h at 37 °C in a mixture of 50 μL Baco IP buffer and 100 μL *p*-nitrophenyl phosphate disodium salt (PNPP). Centrifugation was at 1200 rpm for 5 min, and 100 μL of the end-product was distributed on 96 well plates and analyzed using the ELISA reader at 405 nm of wavelength. 

### 4.8. Immunohistochemistry 

Tumor samples were cryosectioned into 8-μm sections and mounted on Superfrost Plus slides (Menzel-Glaser, Braunschweig, Germany). Tissue slices were subjected to immunohistochemistry against primary antibodies: p-STAT3 (#9134S) (1:200, Cell signal, Danvers, MA, USA) and SHP-2 (SC-7384) (1:200, Santa Cruz, Burlington, MA, USA). The immunoreactive signals were observed under a confocal microscope, with the AF 488-conjugated donkey antimouse IgG and AF 594-conjugated donkey antirabbit antibody (1:200 Invitrogen, Waltham, MA, USA).

### 4.9. Histological Examination

After transcardial perfusion with 4% paraformaldehyde in 0.1 M phosphate buffer (pH 7.4), the weight and size of the subcutaneous tumor were measured. The subcutaneous tumor and Matrigel plug were embedded, longitudinally cut into 8 μm thick sections, and stained with hematoxylin-eosin (H&E), DAP substrate, phosphate-STAT3, and SHP-2.

### 4.10. Zymography

The cultured media of U87turns brown MG cells under different treatments were obtained, and centrifuged at 1200 rpm, 4 °C for 10 min. Supernatant of 30 μg was mixed with a 5 μL sample buffer in 10% SDS-PAGE wells, and then subjected to electrophoresis for 4 h. The gel was washed twice with 25% Triton X-100 in 30 min. The reaction product of 50 mL was incubated at 37 °C overnight and stained with a solution containing 60 mL methanol, 20 mL Glacial acetic acid, and Coomassie brilliant blue. The scanned results were presented in terms of density levels.

### 4.11. Soft Agar Assay

Sterilized agar solutions at 1.4% and 0.7% were melted by microwave and maintained in a water bath at 37 °C. The mixture of 1.4% agar solution and culture medium were used to fill 6-well culture plates. The mixture of 2 × 10^4^ U87 MG cells and 0.7% agar solution (1:1) was added into the above-said 6 wells incubated at 37 °C and in 5% CO_2_ for 21 days, with the medium being changed twice a week. Finally, the 6 wells plates were stained with 0.04% Giemsa solution, and results were presented in terms of the number of colonies.

### 4.12. Wound Healing Assay

In 12-well culture plates, 1 × 10^5^ U87 MG cells/mL was seeded on each well to a final volume of 1 mL until confluent growth had been reached in each well. Scraping the culture well with a 200 μL microtip with a width of 1 mm, different PPAR γ agonists (ciglitazone, troglitazone, and 15d-PGJ2) at μM, photographs were taken on days 0, 1, and 3, and results analyzed. 

### 4.13. Acridine Orange Staining

U87 MG cells treated with 20 μM ciglitazone for 24 h were incubated with 1 μg/mL acridine orange for 15 min. Red color products, representing the positive response, were observed under a confocal microscope. 

### 4.14. Flow Cytochemistry

U87 MG cells seeded in 6 cm culture dishes were treated with ciglitazone. The supernatant was discarded and washed with PBS twice before adding Trypsin-EDTA. The floating cells were washed with PBS followed by −20 °C 70% EtOH before filtration. Results were incubated in a solution with 100 L RNase A (2 μg/mL) and 100 μL Propidium iodide (400 μg/mL) at 37 °C for 30 min. Cell samples finally underwent FACSCalibur flow cytometry and analyzed with the CellQuest. 

### 4.15. MTT Assay

U87 MG cells in a concentration of 1 × 10 ^5^ cells/mL were seeded in 96-well culture plates in a final volume of 100 μL/well. Added to each well was a 10 μL solution containing 5 mg/mL 3-(4,5-Dimethyl-2-thiazolyl)-2,5-diphenyl-2H-tetrazoliumbromide (MTT), and incubation was at 37 °C for 4 h. After discarding the supernatant and adding 100 μL DMSO, each well was analyzed with the ELISA reader at a wavelength of 570 nm.

### 4.16. Statistical Analyses

All data were presented as the mean ± standard error (SE). Intergroup differences were assessed with one–way analysis of variance (ANOVA) followed by Dunnett’s test. Statistical significance was set at *p* < 0.05.

## 5. Conclusions

Ciglitazone inhibits GBM growth mainly through the activation of SHP-2 to impede the formation of phospho-STAT3, as well as activate the proteosome and lysosome to degrade cell-cycle-associated proteins. Ciglitazone is likely a potential candidate in the treatment of GBM.

## Figures and Tables

**Figure 1 ijms-23-13510-f001:**
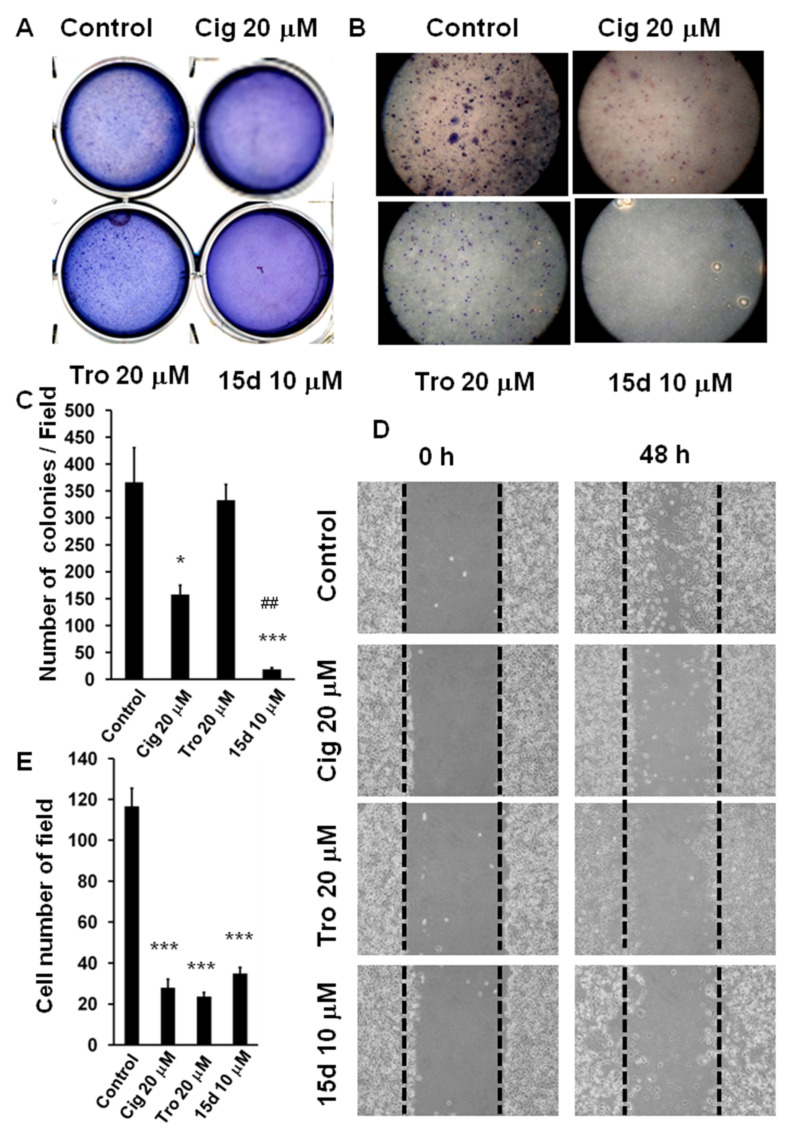
Arrest of glioma tumor growth and migration by PPARγ agonist. U87 were seeded at 2 × 10^4^ cells/mL in soft agar and treated with PPARγ agonists, ciglitazone 20 μM, troglitazone 20 μM, or 15d-PGJ2 10 μM. Colony staining was performed with 0.04% Giemsa solution at day 21. Measurement of U87 cells migration by several PPARγ agonists assessed in wound healing assay with U87 cells seeded at 2 × 10^5^ cells/mL and treated with different PPARγ agonists. (**A**) Representative U87 growth pattern in soft agar treated in different PPARγ agonists. (**B**) Giemsa staining of U87 with Giemsa solution on day 21. Measurement of U87 cells migration by several PPARγ agonists assessed in cell colonies treated in different PPARγ agonists under 100× magnification (**C**) Quantitative analysis of colony formation in different treatment groups repeated with three independent tests. (**D**) The photography of wound healing test treated with different PPARγ agonists including ciglitazon, troglitazone, and 15d-PGJ2 at the 5 μM taken under 100× magnification and repeated in three independent tests. (**E**) Quantitative analysis of number of cells in each field for different treatment groups. *: *p* < 0.05; ***: *p* < 0.001 indicated the experiment relative to control. ##: *p* < 0.01 indicated the experimental group (15d-PGJ2) relative to another experimental group (troglitazone).

**Figure 2 ijms-23-13510-f002:**
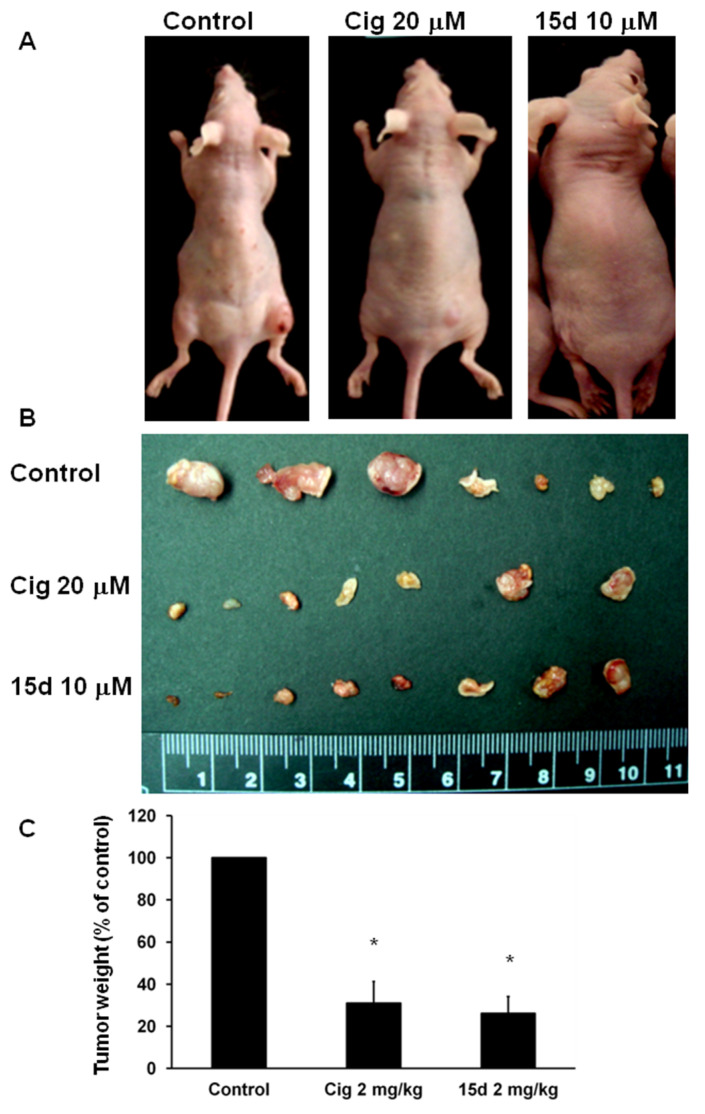
Effect of PPARγ agonists on human glioblastoma cancer cell (U87) tumor xenograft. U87 (5 × 10^6^ cells) were injected subcutaneously at the flank of nude mice. When tumors grew to 5 mm in diameter, the mice were given vehicle (*n* = 6), 2 mg/kg ciglitazone (*n* = 6), or 2 mg/kg 15d-PGJ2 (*n* = 6) twice a week for 10 days. (**A**) Representative photography of tumors in nude mice among different treatment group. (**B**) Photograph of tumor in different treatment group after removal from nude mice. (**C**) The weight of removed tumor in the different treatment group was measured. Data are presented as mean ± standard error (SE). *: *p* < 0.05.

**Figure 3 ijms-23-13510-f003:**
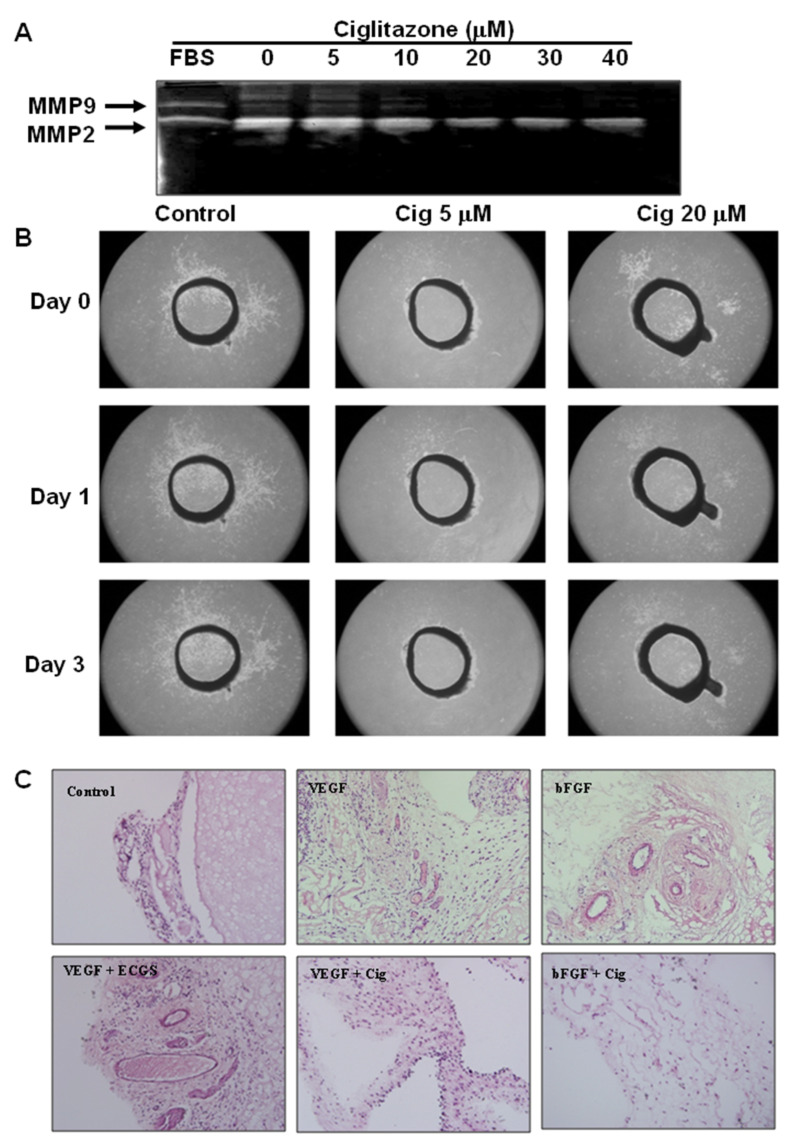
Inhibition of angiogenesis by ciglitazone in vitro and in vivo assay. U87 cells were cultured in serum free media by adding escalated concentration of ciglitazone from 5 to 40 μM. The media were collected and MMP-2 and MMP-9 activities were analyzed by gelatin zymography. Rat aortic rings were also treated with ciglitazone 5 μM, or 20 μM, respectively, and the photography was taken at day 0, day 1, and day 3 under 100× magnification. In the vivo assay, Matrigel plug sections in mice treated were with VEGF+ ECGS, VEGF, VEGF+ ciglitazone, bFGF, and bFGF+ ciglitazone. (**A**) Representative photography in MMP-2 and MMP-9 activities in gelatin zymography. (**B**) Representative photography in endothelium migration treated with ciglitazone 5 μM, or 20 μM related to different time points of day 0, day 1, and day 3 under 100× magnification. (**C**) Representative images of hematoxylin and eosin (HE) stained in Matrigel plug sections in mice treated with VEGF+ ECGS, VEGF, VEGF+ ciglitazone, bFGF, and bFGF+ ciglitazone. Control as a negative control; VEGF+ ECGS as a positive control. The photography was taken under 200× magnifications.

**Figure 4 ijms-23-13510-f004:**
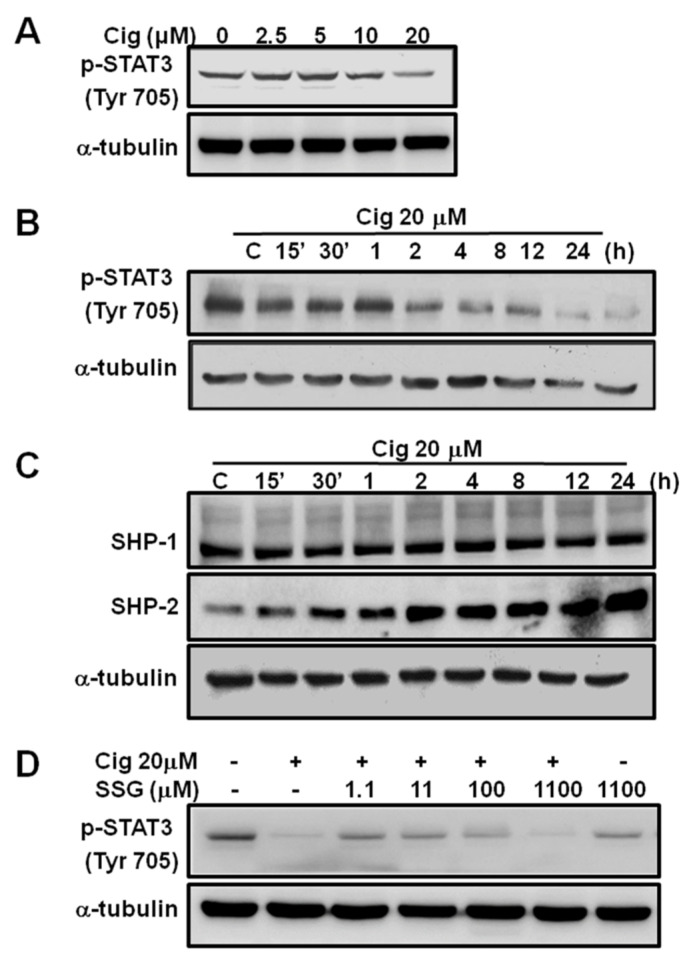
The suppression of phosphorylation of STAT3 at Tyr705 by ciglitazone is highly related to SHP-2 activation. U87 were treated with ciglitazone in a dose-dependent course or time-dependent course and analyzed phospho-STAT3 (Tyr 705), SHP-1, and SHP-2 by Western blotting. (**A**) Representative of expression of P-STAT 3 in the different concentration of ciglitazone harvested 24 h after the treatment. (**B**) Representative of expression in p-STAT3 related to different time profile at the treatment of ciglitazone of 20 μM. (**C**) U87 were treated with ciglitazone in a dose and time dependent manner and then analyzed in SHP-1 and SHP-2 expression by Western blotting. (**D**) U87 were treated with sodium stibogluconate (SSG), a protein tyrosine phosphatase inhibitor, at the indicated concentration for 24 h and analyzed in phospho-STAT3 (Tyr 705) by Western blotting. α-tubulin were loaded as controls.

**Figure 5 ijms-23-13510-f005:**
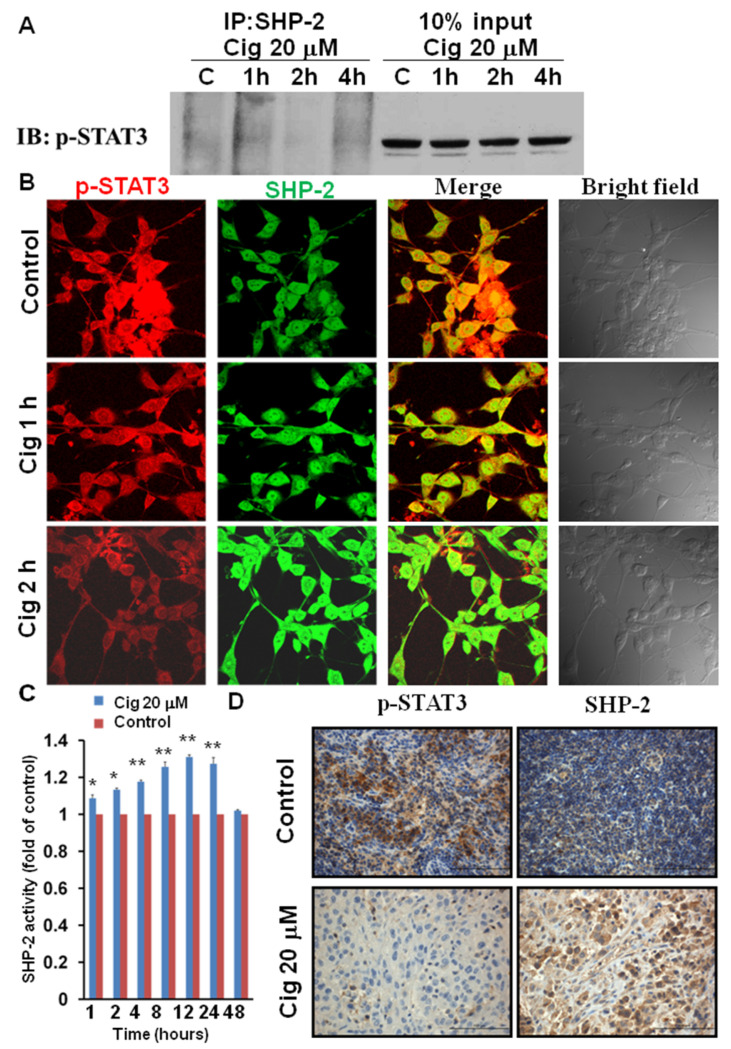
Interaction between phospho-STAT3 and SHP-2 activity by ciglitazone in vitro and vivo study. U87 cells were treated with ciglitazone and the whole cell lysates immunoprecipitated with SHP-2 and immunoblot with phospho-STAT3 (Tyr 705). U87 were treated with vehicle or ciglitazone 20 μM for 1 h and whole cell lysates for SHP-2 activity. Immunohistochemistry staining was used to assess U87 cells and subcutaneous glioma tissue to measure the phospho-STAT3 (Tyr 705) and SHP-2 expression. (**A**) U87 cells were treated with ciglitazone in a time-dependent course and the whole cell lysates immunoprecipitated with SHP-2 and immunoblotted with phospho-STAT3 (Tyr 705). (**B**) Immunocytochemical analysis of SHP-2 (green) and phospho-STAT3 (Tyr 705) (red) proteins was shown under 400× magnification. Colocalization of two proteins was shown as merged image. (**C**) U87 were treated with vehicle or ciglitazone 20 μM for 1 h and whole cell lysates for SHP-2 activity assay determined at the different time profile with three repeated experiments. (**D**) Immunohistochemistry analysis for phospho-STAT3 (Tyr 705) and SHP-2 in subcutaneous glioma tissue using DAB substrate and haematoxylin counterstaining. The photography was taken at 400× magnification. *: *p* < 0.05; **: *p* < 0.01.

**Figure 6 ijms-23-13510-f006:**
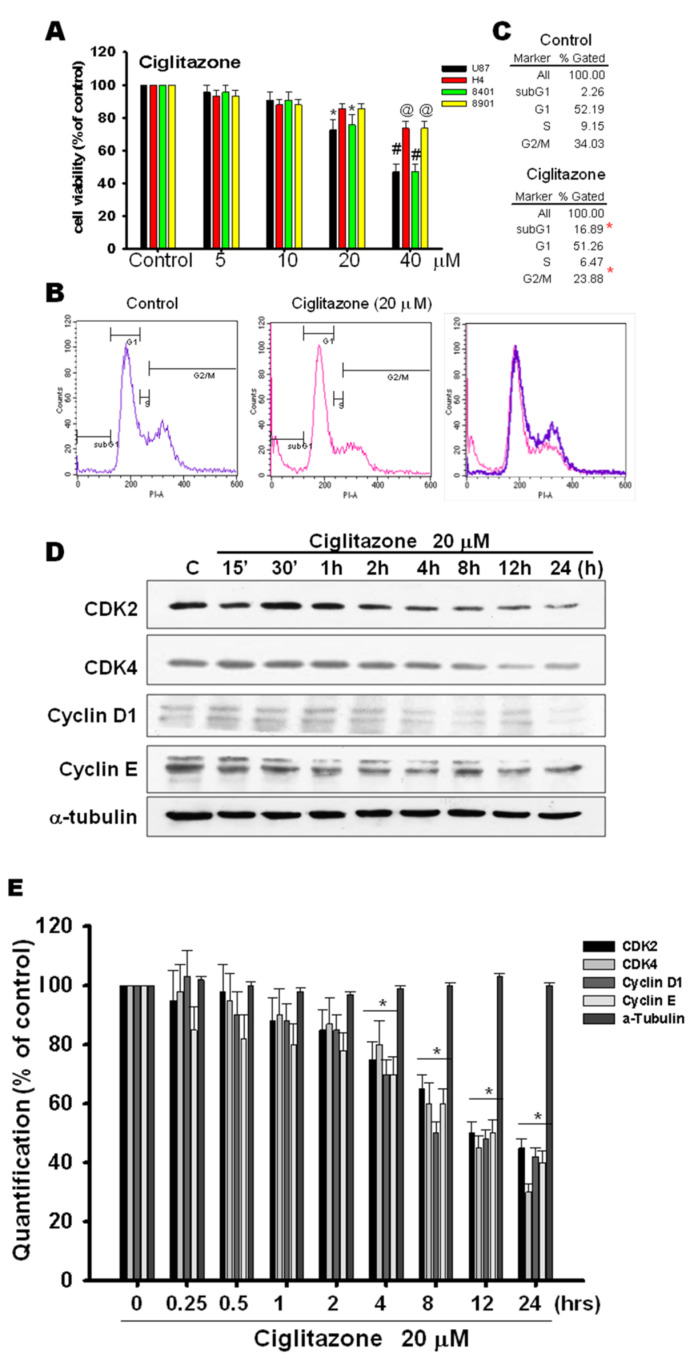
The influence of cell cycle associated proteins by ciglitazone. U87 were treated with ciglitazone in a dose-escalated course and analyzed by MTT and flow cytometry assay and also determined in cell-cycle-associated proteins. (**A**) Four glioma tumor cell lines, U87 (black), H4 (red), 8401 (green), and 8901 (yellow) were analyzed under dose-dependent (0, 5,10, 20, and 40 μM) using MTT assay at 24 h. (**B**) Cell cycle histogram showed phase distribution (subG0, G1, S, and G2/M) of cells at 24 h post-ciglitazone-induction in U87 cells. The plot showed the increase proportion in SubG0 phase but decrease in G2/M phase after ciglitazone treatment. (**C**) Upper panel for control in cell phase distribution, and lower panel for ciglitazone treatment in cell phase distribution. (**D**) Ciglitazone (20 μM) induced cell cycle arrest by Western blot analysis. Western blot analysis showed the decrease of CDK2, CDK4, Cyclin D1, and Cyclin E in time-dependent course in U87 cells. (**E**) Relative quantification (% of control) was evaluated by densitometry for Western blot analysis in CDK2, CDK4, Cyclin D1, Cyclin E, and α-tubulin in time-course-dependent in U87 cells. Data are expressed as mean ± SD (*n* = 6); * *p* or ^@^ *p* < 0.05 vs. control group cells. #: *p* < 0.01 compared with control group. α−tubulin was loaded as a control.

**Figure 7 ijms-23-13510-f007:**
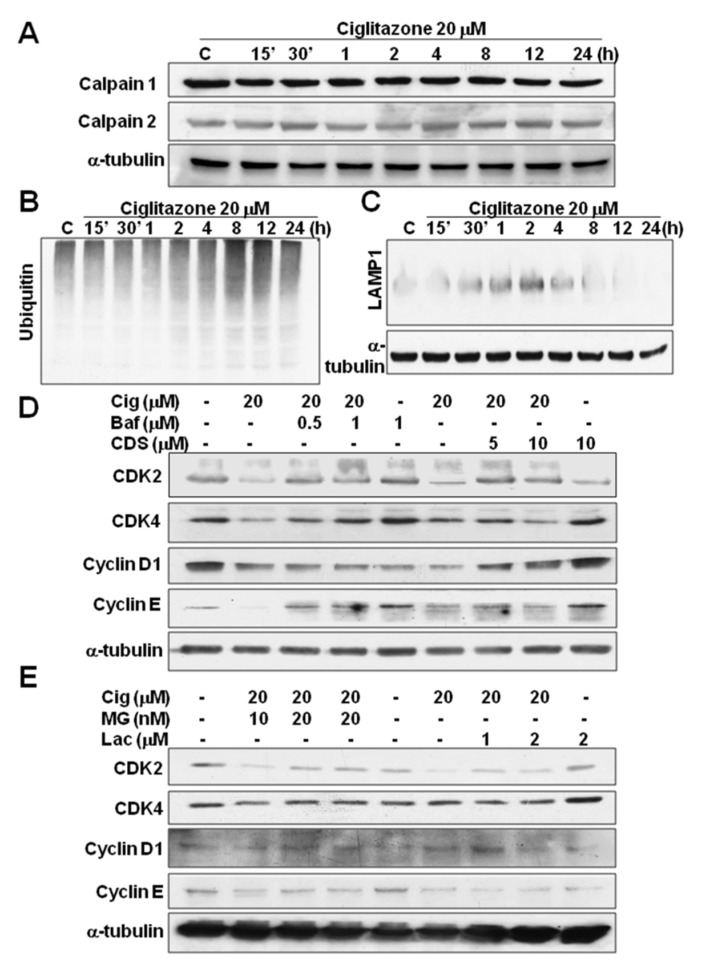
Assessment of cell cycle degradation proteins induced by ciglitazone. U87 were treated with ciglitazone in an escalated time course and analyzed for ER stress marker (calpain I/II), proteasome marker (ubiquitin), and lysosome marker (LAMP1) by Western blotting. U87 cells were also treated with ciglitazone and then by adding proteasome and lysosome inhibitors incubated for 24 h and analyzed for CDK2, CDK4, Cyclin D1, and Cyclin E by Western blotting. (**A**) Determination of calpain I/II in U87 cell line treated with ciglitazone (20 μM) in time-dependent course. (**B**) Determination of ubiquitin in U87 cell line treated with ciglitazone (20 μM) was in a time-dependent course. (**C**) Determination of LAMP1 in the U87 cell line treated with ciglitazone (20 μM) was in a time-dependent course. (**D**) Determination of CDK2, CDK4, Cyclin D1, and Cyclin E in U87 cells treated with ciglitazone subjected to MG132 (MG) and Lactacystin (Lac) treatment. (**E**) Determination of CDK2, CDK4, Cyclin D1, and Cyclin E in U87 cells treated with ciglitazone subjected to Bafilomycin (Baf) and Chloroquine diphosphate salt (CDS) treatment. (**F**) Relative quantification (% of control) and normalization α-tubulin were evaluated in calpain-I and calpain-II. (**G**) Relative quantification (% of control) of Ubiquitin by densitometry in U87 cells treated with 20 μM ciglitazone related to different time profile. (**H**) Relative quantification (% of control) of LAMP1 by densitometry in U87 cells treated with 20 μM ciglitazone related to different time profile. (**I**) Cells pretreatment with Baf and CDS following ciglitazone exposure for protein detection. Relative quantification (% of control) in CDK2, CDK4, Cyclin D1, and Cyclin E protein expression and normalization with α-tubulin were evaluated in U87 cells. (**J**) Cells pretreatment with MG and Lac following ciglitazone exposure for protein detection. Relative quantification (% of control) in CDK2, CDK4, Cyclin D1, and Cyclin E protein expression and normalization with α-tubulin were evaluated in U87 cells. Data are expressed as mean ± SD (*n* = 6); * or # or $: *p* < 0.05 as compared with control. α-tubulin was loaded as a control.

**Figure 8 ijms-23-13510-f008:**
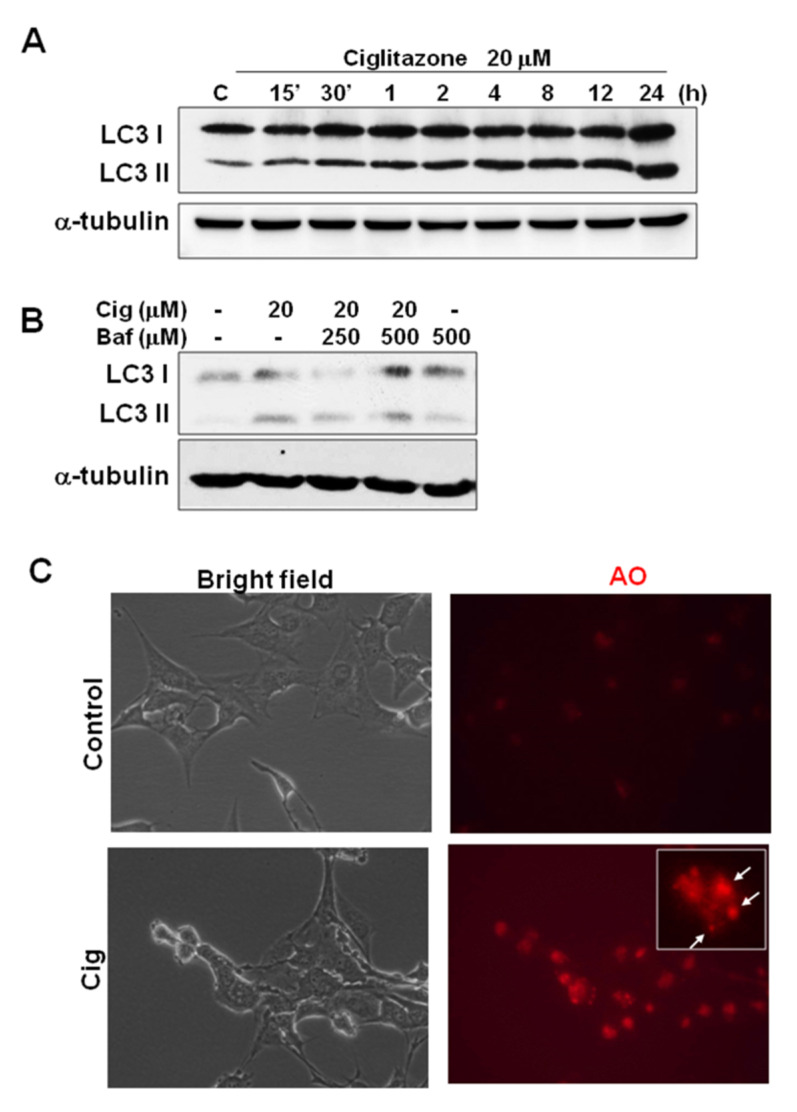
Induction of autophagy marker LC3 and acidic vesicular organelles (AVOs) by ciglitazone in U87 glioma cells. (**A**) U87 were treated with ciglitazone in a time-dependent course and analyzed in LC3 expression by Western blotting. (**B**) The induction of LC3 was affected by bafilomycin. (**C**) The staining AVOs by acridine orange stain were shown in the different treatment group under 400× magnification. (**D**) The relative quantification of LC3 I/II induced by ciglitazone related to different time profile was determined in Western blot analysis. The data showed the expression in a time-dependent course. (**E**) The quantitative analysis of LC3I/II protein expression in U 87 cells pretreated with Baf, following ciglitazone exposure. (**F**) Quantification of AO intensity (% of control) in U87 cells treated with 20 μM of ciglitazone. The results revealed the statistically different increase in AO expression after ciglitazone treatment. Data are expressed as mean ± SD (*n* = 6). * *p* < 0.05: as compared with control. The α-tubulin was loaded as a control.

## Data Availability

Not applicable.

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
