# Peer review of "Potential Therapeutic Effects of Thiazolidinedione on Malignant Glioma"

_ijms, 2022, doi:10.3390/ijms232113510_

Round 1

Reviewer 1 Report

Review of the manuscript entitled: Potential therapeutic effects of thiazolidinedione on malignant glioma. Manuscript is interesting but some corrections should be made.

A clear purpose of the manuscript should be written, e.g. “The aim of the present study was to…” Please add aim to abstract and introduction.

Reference are missing in lines: 40, 61,

Typos, errors, and incorrect symbols in lines: 85, 89, 131, 145, 161, 162, 163, 216, 297, 318-321

Authors should decide how to write STAT 3, or STAT3, or STAT-3 in the manuscript I found all the possibilities.

Figure 6 A, I suggests to present the results as a percentage of the control, the control should be as 100%. 6 C - if possible, please do western blot densitometry. Similarly, figures 7 and 8, please, if possible, do a densitometry.

The discussion is too shallow. The results should be analyzed more deeply. Please refer to the data in the literature and compare it with other publications. If there is no ciglitazone then with other TZDs or other types of cancer.

In the methodology, antibody catalog numbers, and dilutions should be given, is essential.

Author Response

Response to comments

Reviewer 1:

Review of the manuscript entitled: Potential therapeutic effects of thiazolidinedione on malignant glioma. Manuscript is interesting but some corrections should be made.

  1. A clear purpose of the manuscript should be written, e.g. “The aim of the present study was to…” Please add aim to abstract and introduction.

Response to comments: Yes, it is. Thank you for your prestigious comments. We revised our abstract and introduction” ----The aim of this study is to investigate the potential therapeutic effect of PPAR gamma agonist on the malignant glioma. --”

  1. Reference are missing in lines: 40, 61,

Response to comments: Yes, it is. We added two references in line 40 and line 61.The references were shown below:

Ostrom, Q. T.; Cote, D. J.; Ascha, M.; Kruchko, C.; Barnholtz-Sloan, J. S., Adult Glioma Incidence and Survival by Race or Ethnicity in the United States From 2000 to 2014. JAMA Oncol 2018, 4, 1254-1262.

Luwor, R. B.; Stylli, S. S.; Kaye, A. H., The role of Stat3 in glioblastoma multiforme. J Clin Neurosci : official journal of the Neurosurgical Society of Australasia 2013, 20, 907-11.

  1. Typos, errors, and incorrect symbols in lines: 85, 89, 131, 145, 161, 162, 163, 216, 297, 318-321

Response to comments: Thank you for your comments. We corrected all errors marked with red color.   

  1. Authors should decide how to write STAT 3, or STAT3, or STAT-3 in the manuscript I found all the possibilities.

Response to comments: We corrected the difference and changed to STAT3. 

  1. Figure 6 A, I suggests to present the results as a percentage of the control, the control should be as 100%. 6 C - if possible, please do western blot densitometry. Similarly, figures 7 and 8, please, if possible, do a densitometry.

Response to comments: Thank you for your suggestion. We revised Figure 6A, C and add the statistical analysis in Figure 6, 7, and 8. 

  1. The discussion is too shallow. The results should be analyzed more deeply. Please refer to the data in the literature and compare it with other publications. If there is no ciglitazone then with other TZDs or other types of cancer.

Response to comments: Thank you for your comments. We revised the discussion section.

  1. In the methodology, antibody catalog numbers, and dilutions should be given, is essential.

Response to comments: Thank you for your comments; we add the data catalog numbers and dilution in the supplementary table I.

Reviewer 2 Report

The authors have investigated thiazolidinedione analogs (mainly Ciglitazone) as a potential treatment for Glioblastoma.  They have conducted some nice experiments but there are many problems.

Minor:

The manuscript needs to be edited by a native English speaker. Throughout the manuscript the micro symbol (mu) is replaced with a spiral character.  In many of the figures the times above the lanes don't line up with the correct lanes.  

This manuscript is about thiazolidinedione analogs but nowhere do they mention that Ciglitazone is a thiazolidinedione analog - this should be mentioned in the introduction.

The Figure legends should describe what was done in each panel in more detail. The reader should not have to go to the text to find this - the text should provide even more detail. 

line 96: "All PPARγ have the significant power..." should be "All PPARγ agonists have significant power..."

The sentence on lines 159 and 160 should have a reference.

Line 216: GAPDH and  - tubulin (the alpha is replaced by the spiral character).  Where are these data?

Lines 226-7: It is well known - provide a reference.

Fig. 4D: The SSG concentration over the 5th lane should be 110 not 10.

Fig. 6B: there should be key to define the red and blue lines - this isn't even mentioned in the legend.

Fig. 7E: For the MG row, I think the numbers are shifted - they are not consistent with the other inhibitors.

Line 271 states that Cit did not change cell cycle (which is not true) but line 289 states that it did.

In the Methods, the discussion of IP (line 375) only discusses the immunoblotting not the IP.  The Zymography discussion (line 419) is very hard to understand.

Patents (line 466) should be described (or the authors left out the "no" for no patents.

More major: 

Fig. 3. The authors show VEGF + ECGS but not with agonist: why not?

Line 241 states that protein degradation was mainly through ER stress etc. but line 249 says it is not (which is somewhat consistent with the data) - I think the first statement is a generalization but the authors don't make that clear.  However, the data in Fig. 7A show that Calpain I is increased at 8 and 12 h, contrary to what the authors state.

The authors should look at the cell cycle directly by flow cytometry. Their data with the cyclins are ambiguous.  The use of bafilomycin as a marker for both lysosomal activity and autophagy (even though autophagy occurs through the lysosome) may confuse the reader.

Fig 7 D and E: Unless I am missing something lanes 2 and 6 should be the same but they are not.  The data in the gels is inconsistent and does not confirm what the authors are saying.  The alpha tubulin in the last 2 lanes of E are much lower, further confounding the data.

The subcutaneous tumor models are nice but what happens with orthotopic tumors?  Does ciglitazone increase lifespan?

Author Response

Response to comments

Reviewer 2:

The authors have investigated thiazolidinedione analogs (mainly Ciglitazone) as a potential treatment for Glioblastoma.  They have conducted some nice experiments but there are many problems.

 Minor:

  1. The manuscript needs to be edited by a native English speaker. Throughout the manuscript the micro symbol (mu) is replaced with a spiral character.  In many of the figures the times above the lanes don't line up with the correct lanes.  

Response to comments: Thank you for your comments. The co-author of Dr. Jason Sheehan, who is the professor in the department of neurosurgery, makes the final editing.

  1. This manuscript is about thiazolidinedione analogs but nowhere do they mention that Ciglitazone is a thiazolidinedione analog - this should be mentioned in the introduction.

Response to comment: Thank you for your comments. We revised the essential information in the text.

  1. The Figure legends should describe what was done in each panel in more detail. The reader should not have to go to the text to find this - the text should provide even more detail. 

Response to comments: Thank you for your comments. We revised our figure legends to make it more detail.

  1. line 96: "All PPARγ have the significant power..." should be "All PPARγ agonists have significant power..."

Response to comments: Yes, it is. We revised the description” The above-said PPARγ have the significant power to inhibit U87 cell migration (Figure 1D, E).” 

  1. The sentence on lines 159 and 160 should have a reference.

Response to comments: Yes, it is. We added a new reference as shown below:

Kim, J. E.; Patel, M.; Ruzevick, J.; Jackson, C. M.; Lim, M., STAT3 Activation in Glioblastoma: Biochemical and Therapeutic Implications. Cancers 2014, 6, 376-95.

  1. Line 216: GAPDH and  α- tubulin (the alpha is replaced by the spiral character).  Where are these data?

Response to comments: Yes, it is. We revised the data and remove them.

  1. Lines 226-7: It is well known - provide a reference.

Response to comments; Yes, it is. We added a new reference as below:

He, G.; Thuillier, P.; Fischer, S. M., Troglitazone inhibits cyclin D1 expression and cell cycling independently of PPARgamma in normal mouse skin keratinocytes. The J Invest Dermatol 2004, 123, 1110-9.

  1. 4D: The SSG concentration over the 5th lane should be 110 not 10.

Response to comments: Yes, it is. We revised the mistake.

  1. 6B: there should be key to define the red and blue lines - this isn't even mentioned in the legend.

Response to comments: Yes, it is. We revised the figure and legend.

  1. 7E: For the MG row, I think the numbers are shifted - they are not consistent with the other inhibitors.

Response to comments: Yes, it is. We edited Figure 7E.  

  1. Line 271 states that Cit did not change cell cycle (which is not true) but line 289 states that it did.

Response to comments: Yes, it is. We revised the text and made it much clearer.

  1. In the Methods, the discussion of IP (line 375) only discusses the immunoblotting not the IP.  The Zymography discussion (line 419) is very hard to understand.

Response to comments: Yes, it is. We revised the data and re-edited the figures.

  1. Patents (line 466) should be described or the authors left out the "no" for no patents.

Response to comments: Yes, it is. We revised the description.

More major: 

  1. 3. The authors show VEGF + ECGS but not with agonist: why not?

Response to comments: Yes, it is.  The control group was regarded as a negative control, but VEGF + ECGS as a positive control, which indicated the greatest vessels lumen in the sissue..  

  1. Line 241 states that protein degradation was mainly through ER stress etc. but line 249 says it is not (which is somewhat consistent with the data) - I think the first statement is a generalization but the authors don't make that clear.  However, the data in Fig. 7A show that Calpain I is increased at 8 and 12 h, contrary to what the authors state.

Response to comments: Yes, it is. We repeated our experiment and made them clearer. We also repeated our experimental and re-edited Figure 7A.    

  1. The authors should look at the cell cycle directly by flow cytometry. Their data with the cyclins are ambiguous.  The use of bafilomycin as a marker for both lysosomal activity and autophagy (even though autophagy occurs through the lysosome) may confuse the reader.

Response to comments: Yes, it is. We repeated our experiment and re-edited our data and text.

  1. Fig 7 D and E: Unless I am missing something lanes 2 and 6 should be the same but they are not.  The data in the gels is inconsistent and does not confirm what the authors are saying.  The alpha tubulin in the last 2 lanes of E are much lower, further confounding the data.

Response to comments: Yes, it is. We revised the figures and add the statistical analysis of densitometry data in western blot.  

  1. The subcutaneous tumor models are nice but what happens with orthotopic tumors?  Does ciglitazone increase lifespan?

Response to comments: Yes, it is. The orthotopic tumor model is an essential model to investigate the tumor growth and animal survival. But the subcutaneous tumor model is an easy way to investigate the tumor growth observed by photography. In the following study, the orthotopic use should be considered.    

Round 2

Reviewer 1 Report

the manuscript was corrected

Author Response

Thank you for your recommendation. 

Reviewer 2 Report

The authors made some of the requested changes but there are still many careless mistakes. The English grammar of manuscript is still rather poor and although the authors state that Dr. Sheehan reviewed the grammar, it appears that he did not. 

Line 82: (thiazolidinedione analogs) should be (a thiazolidinedione analog).

Line 99: PPARy agonists

Fig. 4D, lane 5: the authors changed it from 10 but it should be 110 not 100.

Fig. 6B: state the percentage of cells in each phase of the cell cycle.

Fig. 7 (figure legend, lien 298): the authors state that *, # and @ are used to indicate p<0.05. Why use different symbols - this is confusing.  The also have & in the graph but do not say what is means.

Line 305: Should be quantitation not qunatitation.

Although the authors state that they made the requested changes to the Methods, they did not.  They do not describe how the IPs were conducted  - only the immunoblotting after the IP.  The first line of the Zymography section (line 455) does not make sense. 

The authors need to provide details on how the cell cycle analyses were done with flow cytometery.

Author Response

  1. Line 82: (thiazolidinedione analogs) should be (a thiazolidinedione analog).

Response to comments: Yes, it is. We revised line 82 as following “Ciglitazone (a thiazolidinedione analog) is a well-known PPARγ ligand in treating diabetes with the potential for tumor treatment [24, 25].”

  1. Line 99: PPARy agonists

Response to comments: Yes, it is. We revised text in line 99” The above-said PPARγ agonists have the significant power to inhibit U87 cell migration (Figure 1D, E).”

  1. 4D, lane 5: the authors changed it from 10 but it should be 110 not 100.

Response to comments: Yes, it is. It should be 110 and we made the change of 110 in lane 5.

  1. 6B: state the percentage of cells in each phase of the cell cycle.

Response to comments: We rewrote the Figure 6B legend ” (B) Cell cycle histogram showed phase distribution (subG1, G1, S and G2/M) of cells at 24 hours post ciglitazone-induction in U87 cells. The histogram showed the increased proportion of subG1 phase but decreased proportion in G2/M phase.”

  1. 7 (figure legend, lien 298): the authors state that *, # and @ are used to indicate p<0.05. Why use different symbols - this is confusing. The also have & in the graph but do not say what is means.

Response to comments: We revised the Figure 7 and legends” *: P<0.05 and #: P<0.01 as compared with control; $: p<0.05 as compared with the group with 20ug ciglitazone.”

  1. Line 305: Should be quantitation not qunatitation.

Response to comments: Yes, it is. We corrected the mistake in line 305” The quantitative analysis was shown in Figure D-E.”

  1. Although the authors state that they made the requested changes to the Methods, they did not. They do not describe how the IPs were conducted- only the immunoblotting after the IP.  The first line of the Zymography section (line 455) does not make sense.

Response to comments: We made the revision of methodology in immunoprecipitation and zymography:”

Immunoprecipitation analyses:

Protein pre-clean (800 mg) was incubated with preimmune serum for 1 h at 4ºC with gentle agitation. Supernatant further were incubated with specific antibodies and immobilized immunoprecipitated with protein A-Sepharose for overnight at 4ºC. Beads were pelleted by centrifugation at 2,500 x g, washed three times with IP buffer, and analyzed by electrophoresis and immunoblot as it was indicated above. Proteins were separated by sodium dodecyl sulfate – polyacrylamide gel electrophoresis, and electrophoretically transferred to nitrocellulose membranes. ------

Zymography

The cultured media from U87 cells (1x106 cells) under different treatments were obtained, centrifuged at 1,200 rpm, 4 °C for 10 minutes and were subjected to electrophoresis under non-reducing conditions using either 10 % gelatin type B containing gels.--------“

  1. The authors need to provide details on how the cell cycle analyses were done with flow cytometery.

Response to comments: Yes, it is. We revised the methodology” 

Flow cytochemistry:

U87 MG cells seeded in 6 cm culture dishes were treated with ciglitazone or 0.1% DMSO for 24 h. The supernatant was discarded and washed with PBS twice, before adding Trypsin-EDTA. The floating cells were washed with PBS followed by -20℃ 70% EtOH at 4°C overnight before filtration. Results were incubated in a solution with 100 L RNase A (2 μg/mL) and 100 μL Propidium iodide (400 μg/mL) at 37 ℃ for 30 minutes in the dark and subjected to a flow cytometric analysis to determine the percentage of cells at specific phases of the cell cycle. Samples also were analyzed in a flow cytometer using lasers that can detect serial of information about cell. Detectors aimed directly in line with a single laser beam (forward scatter, FSC) determine its cell size while detectors aimed perpendicular to the laser beam (side scatter, SSC) assess granularity within the cytoplasm of cells. A flow cytometric analysis was performed using a FACSCalibur flow cytometer (Becton Dickinson, San Jose, CA) equipped with a 488 nm argon laser. The events were evaluated for each sample and the cell cycle distribution was analyzed using Cell Quest software (Becton Dickinson). The results were presented as the number of cells versus the amount of DNA, as indicated by the intensity of a fluorescence signal. All the experiments were conducted four times.